*Cambridge Prisms: Global
Mental Health*

and Song D-H (2025). Effects of parenting
attitudes on the suicide risk of adolescents in
South Korea and the moderating effect of out-
of-school youth status. *Cambridge Prisms:
Global Mental Health*, **12**, e48, 1–8

parenting attitude; suicide risk; adolescent;
out-of-school youth; moderating effect

**Corresponding author:**
Sung-Hee Lee;
Email: s.lee@derby.ac.uk

# Effects of parenting attitudes on the suicide risk of adolescents in South Korea and the moderating effect of out-of-school youth status

Kyu-Hyoung Jeong[1], Sung-Hee Lee[2], A-Ran Park[1] and Do-Hun Song[1]

[1]Department of Social Welfare, Jeonbuk National University, Jeonju, South Korea and [2]Department of Criminology and
Social Sciences, University of Derby, Derby, UK

## Abstract

This study aimed to investigate the impact of parenting attitudes on the suicide risk of
adolescents in South Korea, and to verify the moderating effect of out-of-school youth status.
Utilizing data from the Mental Health Survey of Korean Adolescents (2021), conducted by the
National Youth Policy Institute, 5,937 school youths (SYs) and 752 out-of-school youths (OSYs)
were selected for this study. Multiple regression analysis was used to analyze the relationship
between parenting attitudes and the impact on the suicide risk of adolescents along with the
moderating effect of OSY status. Parenting attitudes consist of a total of six sub-types: warmth,
autonomy support, structure, rejection, coercion and chaos. The results showed that the
parenting attitude of warmth, autonomy support and rejection had a significant effect on the
risk of suicide among adolescents. The study also confirmed that OSY had a moderating effect
related to the parenting attitude types of structure, rejection and chaos. The result of this study
filled the gap in prior research which overlooked the moderating factor of OSY related to parent
attitudes and the suicide risk of adolescents. Some useful insights for practical and policy
measures to reduce the suicide risk of adolescents are suggested.

## Impact statement

This study addresses a critical issue in South Korea, where adolescent suicide rates are alarmingly
high, especially among out-of-school youths (OSYs). By investigating the relationship between
parenting attitudes and suicide risk among adolescents, the research provides valuable insights
into the understanding of how different parenting styles influence suicidal tendencies. Further-
more, the study identifies the moderating role of OSY status, highlighting that OSYs are more
vulnerable to negative parenting behaviors such as rejection and chaos, which significantly
increase their suicide risk. On the other hand, positive parenting behaviors like structure can
mitigate suicide risk more effectively among OSYs compared to school youths (SYs). We believe
that these findings can contribute to the development of targeted interventions for both groups,
suggesting that differentiated parenting education programs should be implemented to address
the specialized and unique needs of OSYs. We also address that this study can support policy
development aimed at providing mental health resources and crisis intervention, especially for
those adolescents with a high risk of suicide, particularly OSYs, as they often have insufficient
access to institutional support systems.

## Introduction

Suicide among adolescents is one of the major social problems worldwide. According to Keeley
(2021), suicide is one of the top five causes of death among adolescents, and almost 46,000
adolescents die from suicide worldwide every year. South Korea (henceforth 'Korea'), the country
with the highest rate of suicide among the Organization for Economic Cooperation and
Development (OECD) member nations, shows a worrisome development in the problem of
adolescent suicide. According to the Ministry of Health and Welfare (2023), as of 2020, the mean
rate of adolescent suicide mortality (per 100,000 population) in OECD member countries is 6.0,
while that of Korea is 11.7, which is almost double the OECD average, and ranks 3rd among
OECD member countries in adolescent suicide rate. In addition, for the last 10 consecutive years,
suicide has been the number one cause of death among Korean adolescents, and the number of
adolescents who took their own lives has been steadily increasing since 2017 (Department for
Women and Families, 2022), clearly showing the grave reality of suicide among adolescents.

Adolescence is a stage of human development in which the youths undergo psychological
and physical transition, and form and develop emotional stability and self-identity; adolescents
are easily exposed to various personal/family/environmental stressors, and managing and

maintaining a mentally healthy state poses a challenge in this period (Choi and Cho, 2016). During adolescence, which is a period of cognitive immaturity and emotional instability before maturing into adulthood, suicide is often viewed as a means of escaping from the difficulties or tribulation at hand rather than a means of abandoning the will to live, making the adolescents susceptible to impulsive suicidal ideation or behavior (Lee et al., 2016). These characteristics of adolescents indicate that proactive discussions and measures are needed to prevent the suicide of adolescents, which comprised the motivation for conducting this study. In particular, this study focused on the teenage years, when mental illness is reported to be concentrated, with those aged 10 to 19 being defined as adolescents and selected as research subjects (National Human Rights Commission, 2018).

Parents are responsible for raising adolescents as they grow into adult members of society and are the ones who have the most direct and significant influence on them. Therefore, the parenting attitude of these parents, which encompasses the general attitude, behaviour and thinking that parents adopt while raising their children, serves as a predictor of delinquent behaviours of adolescents (Kim and Han, 2014). This influence also applies to adolescents' suicide risk. Per the reports of relevant previous studies, parental abuse and neglect were associated with suicidal ideation in adolescents (Nilsen and Conner, 2003; Nelson and Galas, 2006; Draper et al., 2008; Swogger et al., 2011; Jeong, 2021), and in the case of adolescents lacking support from their parents, their relationship with parents had a direct impact on their suicidal ideation and suicidal behaviour (Lewinsohn et al., 2001; Bostik and Everall, 2007; Woo et al., 2010). In addition, adolescents who seriously thought about suicide attempts had lower parental support and more severe parent–child conflict than those who did not have such suicidal thoughts (Wright, 1985); moreover, physical and sexual childhood abuse from parents was reported to be closely related to the suicide of adolescents (Esposito and Clum, 2002). These results of previous studies indicate that the parenting attitude of the youths' parents is a major predictor of suicide risk in adolescents.

From the perspective of encouraging the motivation of adolescents, Skinner et al. (2005) categorized the parenting attitudes into six sub-types: 'warmth, autonomy support, structure, rejection, coercion and chaos'. Warmth refers to the genuine caring, consideration and emotional support given to the child by the parents. Rejection is the conceptual opposite of warmth and refers to parents showing indifference, hostility/aversion and overt communication of negative feelings such as criticism, derision and irritability toward their children. Autonomy support refers to primary caregivers or parents helping and supporting children in solving problems on their own and making decisions and choices. Contrary to autonomy support, coercion shows key features like authoritarian and autocratic parenting styles and refers to suppressing the voices of children, involving restrictive overcontrolling with rules and expectations set by parents. Structure refers to the provision of clear goals, rules and expectations and resources, including information such as the predictability of desired or undesired outcomes and feedback; at home, the primary caregivers or parents play the role of leaders. Chaos is the conceptual opposite of structure in which parents show inconsistent behaviour and standards to their children depending on their moods or situations. In addition, according to Self-Determination Theory (SDT), adolescence is a period in which three psychological needs, namely autonomy, competence and relatedness, are satisfied and parental parenting attitudes affect the satisfaction of these needs (Assor et al., 2004). Positive parenting factors have a positive effect on the psychological

development of adolescents by satisfying these needs, while negative parenting attitudes are reported to significantly increase their emotional instability (Vansteenkiste & Ryan, 2013; Otterpohl et al., 2019). As can be seen from these six dimensions, parenting attitude is composed of conceptually opposite dimensions, but in terms of relations among these dimensions, trends show that warmth and rejection, which are opposing concepts, are both low or both high at the same time (Skinner et al., 2005), which points out to the need for comprehensive considerations of both positive parenting factors (warmth, autonomy support and structure) and negative parenting factors (rejection, coercion and chaos). However, existing literature investigating the relationship between parenting attitude and the suicide risk of adolescents has the limitation of not including such a comprehensive analysis (Donath et al., 2014; Joung & Seo, 2014; Kim and Kim, 2020; Darvishi et al., 2023). This study, therefore, aimed to examine parenting attitudes as a predictor of suicide risk among adolescents by categorizing them into six sub-types; warmth, autonomy support, structure, rejection, coercion and chaos.

In addition, gender (Beautrais et al., 2006; Jiang et al., 2010), age (Lee, 2018; Jeong, 2021) and household income level (Parker et al., 1997), which have been reported to influence adolescent suicide risk, were included as control variables to provide a clear understanding of the relationship between parenting attitude and adolescent suicide risk.

Out-of-school youths (OSYs) have a higher risk of suicide compared to school youths (SYs) as they are more likely to experience drug involvement, family strain, emotional distress and exposure to violence (Thompson et al., 1994; Thompson and Eggert, 1999), and they have been reported to have higher frequency of suicidal ideation and suicide attempts (Daniel et al., 2006; Jeong, Park, & Kim, 2021). Furthermore, for school-age adolescents, school is generally the place where they spend most of their time during the day, and along with academic knowledge, they acquire skills of socialization through interpersonal relationships with different people, which cannot be learned at home (Shaffer, 2009). As a result, it is a place where the youths are greatly influenced by relationships with other people beyond their relationship with parents (Lee, 2008). However, OSYs spend relatively more time at home and are more influenced by their parents (Park & Yun, 2023). Considering these aspects of OSYs, it is expected that OSY status will have a moderating role between parenting attitudes and the suicide risk of adolescents. However, previous studies related to adolescent suicide have mainly looked into cases of adolescents attending school (Donath et al., 2014; Joung & Seo, 2014; Kim and Kim, 2020; Park & Yun, 2023; Darvishi et al., 2023). Moreover, previous studies have only included OSYs as participants (Daniel et al., 2006; Szlyk, 2020; Jo et al., 2023; Lee and Lee, 2023); even if they included both SYs and OSYs, they performed a simple comparison of how individual factors affect the suicide risk of adolescents (Jeong et al., 2021; Kim, 2021). Thus, there have been few previous studies that examined and verified the changes in suicide risk according to the SY/OSY status of adolescents.

This study aims to analyse the effect of parental parenting attitudes on the suicide risk of adolescents and verify whether out-of-school youth (OSY) moderate this relationship. To this end, parental parenting attitudes were classified into six types: warmth, autonomy support, structure provision, rejection, coercion and inconsistency. The effect on the suicide risk of adolescents was evaluated, and the differences between school adolescents and out-of-school adolescents were analyzed. The significance of this study is that it proposes practical intervention measures to reduce the suicide risk of adolescents and provides basic data for parent education programmes and policy support.

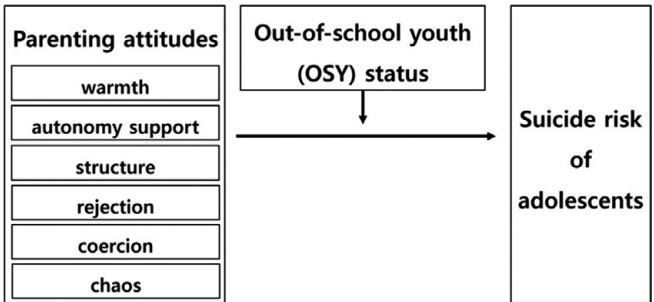

**Figure 1.** The study model.

## Data and methods

### Research model

In this study, we aimed to investigate how parenting attitude affects the suicide risk of adolescents and examine whether OSY status has a moderating effect in this relationship. Figure 1 below shows the research model for this study including variables.

### Data

The Mental Health Survey of Korean Adolescents (2021), collected by the National Youth Policy Institute, was used for this study. The data were aimed at collecting basic information on the major mental health problems experienced by adolescents in Korea and providing evidence for the development of government policies and programs related to the mental health of Korean adolescents. The original sample consisted of a total of 6,689 adolescents, including 5,937 school youths (SYs); 2,039 upper grades of elementary school students, 1,948 middle school students and 1,950 high school students, and 752 OSYs. The SYs were extracted by stratified cluster sampling, and OSYs by convenience sampling. This study was a cross-sectional study that utilized data collected at a specific time. That is, the survey was conducted for the period of approximately one month from July to August 2021 and by means of a self-administered online survey. All research participants were given the consent form for the data collection and use of personal information, with only those adolescents who agreed to all consent, matters being included in this study. In total, 5,937 SYs and 752 OSYs were selected as respondents for final analysis using the data of the Mental Health Survey of Korean Adolescents.

### Variables

#### Dependent variable: suicide risk

The dependent variable in this study was suicide risk, and the Mental Health Screening Tool for Suicide Risk (MHS:S) developed by Yoon et al. (2020) was used as the tool for screening and assessment. The MHS:S consists of four questions on a five-point scale (never = 0, slightly true = 1, true = 2, fairly true = 3, very true = 4). In this study, the average score of the four questions was used, and the higher the score, the higher the suicide risk. Cronbach's α of the suicide risk screening tool was .862.

#### Independent variables: parenting attitudes

The independent variables in this study were the types of parenting attitudes, and the Korean version of the Parents as Social Context Questionnaire for Adolescents (PSCQ_A), a scale developed by Skinner et al. (2005) with the application of a motivational model,

was used, which was translated, adapted and validated for Korean adolescents by Kim and Lee (2017). Parenting attitudes consists of a total of six types; warmth, autonomy support, structure, rejection, coercion and chaos, with 24 questions in total. Among these dimensions, higher scores for warmth, autonomy support and structure indicate a higher level of positive parenting attitude, whereas higher scores for rejection, coercion and chaos indicate a higher level of negative parenting attitude. Parenting attitude was assessed on a four-point scale (Never = 1, Rarely = 2, Somewhat = 3, Very much = 4). For parenting attitudes, the average score for each dimension was used. As for the scale, Cronbach's α for parenting attitudes was .784, and by dimension, Cronbach's α was .940 for warmth, .922 for autonomy support, .856 for structure, .794 for rejection, .806 for coercion and .830 for chaos.

#### Moderator variable: out-of-school youth status

The moderator variable was OSY status, with SY set to 0 and OSY set to 1.

#### Control variables

The control variables used in this study are gender (male = 0, female = 1), age (middle school age or under = 0, high school age = 1), and household income level (set from 1 to 7 points, Very low = 1, Average = 4, Very high = 7).

### Statistical analysis

For data analysis, Stata 15.0 SE was used. Frequency analysis was conducted for analysis of the sociodemographic characteristics of the participants. An independent *t*-test was conducted to test the differences in key variables according to OSY status. Through multiple regression analysis, the impact of parenting attitude on the suicide risk of adolescents and the moderating effect of OSY status were assessed.

## Results

### Sociodemographic characteristics of participants

Upon examining the socio-demographic characteristics of the participants, 3,343 (50.0%) adolescents were male and 3,346 (50.0%) were female; by age, 4,060 (60.7%) respondents were in the middle school age or under, and 2,629 (39.3%) were in the high school age, confirming that there were more respondents who were in the middle school age or under than in the high school age. By household income level, those who answered 'average' accounted for the highest number at 2,989 respondents (44.7%), and there were more adolescents with above-average household income levels than those with below-average household income levels (Table 1).

### Differences in key variables by OSY status

For testing the differences in key variables depending on the OSY status, an independent t-test was performed (see Table 2). The differences were revealed as follows: the student adolescents were shown to have warmth (M = 3.44, SD = .66), autonomy support (M = 3.42, SD = .64), structure provision (M = 3.07, SD = .68), rejection (M = 1.54, SD = .62), coercion (M = 1.90, SD = .69), inconsistency (M = 1.78, SD = .68) and suicidal tendencies (M = .13, SD = .42). The out-of-school adolescents were found to have warmth (M = 3.15, SD = .80), autonomy support (M = 3.19, SD = .78), structure provision (M = 2.83, SD = .80), rejection

**Table 1.** Sociodemographic characteristics of participants (*N* = 6,689)

| Variables | | *N* | % |
|---|---|---|---|
| Gender | Male | 3,343 | 50.0 |
| | Female | 3,346 | 50.0 |
| Age | Middle school age or under | 4,060 | 60.7 |
| | High school age | 2,629 | 39.3 |
| Household income level | 1 Very low | 32 | 0.5 |
| | 2 | 174 | 2.6 |
| | 3 | 528 | 7.9 |
| | 4 Average | 2,989 | 44.7 |
| | 5 | 1,620 | 24.2 |
| | 6 | 890 | 13.3 |
| | 7 Very high | 457 | 6.8 |

*Note*: *N* = numbers, % = percentage.

(M = 1.71, SD = .73), coercion (M = 1.99, SD = .77), inconsistency (M = 1.96, SD = .78) and suicidal tendencies (M = .46, SD = .87). The analysis results showed that the parenting attitude types of warmth ($p < .001$), autonomy support ($p < .001$), structure ($p < .001$), rejection ($p < .001$), coercion ($p < .01$), chaos ($p < .001$) and suicide risk ($p < .001$) all showed statistically significant differences. The parenting attitude types of warmth, autonomy support and structure were significantly higher in SYs than in OSYs, whereas the types of rejection, coercion, and chaos were significantly higher in OSYs than in SYs. The suicide risk was also significantly higher in OSYs than in SYs.

## Model analysis

The results of the analysis on the moderating effect of OSY status in the relationship between parenting attitude and suicide risk of adolescents are presented in Table 3. The power for suicide risk was 16.2% ($R^2$=.162), and the regression equation was statistically significant ($F$ = 80.710, $p < .001$). As a result of analysing the relationship between key variables, among the control variables, gender (Coef. = .082, $p < .001$) had a significant effect on suicide risk. That is, the suicide risk was significantly higher in women than in men.

**Table 3.** Impact of parenting attitude of adolescents on the suicide risk and moderating effect of OSY status

| Variables | | Coef. | S.E. |
|---|---|---|---|
| Constant | | .357 | .058 |
| Control variable | Gender(ref. male) | .082*** | .011 |
| | Age (ref. middle school age or under) | −.013 | .013 |
| | Household income level | −.009 | .005 |
| Independent variables | Warmth (A) | −.066*** | .015 |
| | Autonomy support (B) | −.043** | .016 |
| | Structure (C) | −.019 | .012 |
| | Rejection (D) | .104*** | .012 |
| | Coercion (E) | .014 | .012 |
| | Chaos (F) | .018 | .012 |
| Moderator variable | OSY status (G) | .099 | .128 |
| Interactions | A × G | .065 | .041 |
| | B × G | −.050 | .043 |
| | C × G | −.108** | .033 |
| | D × G | .152*** | .032 |
| | E × G | −.014 | .034 |
| | F × G | .099** | .033 |
| $R^2$ | | .162 | |
| F (sig.) | | 80.710*** | |

**p < .01,
***p < .001.

Among independent variables that included dimensions of parenting attitude, warmth (Coef. = −.066, *p < .001*), autonomy support (Coef. = −.043, *p < .01*) and rejection (Coef. = .104, *p < .001*) were shown to have a significant effect on the suicide risk of adolescents. That is, with lower warmth and autonomy support or with higher rejection, the suicide risk increased. On the other hand, structure, coercion and chaos involved the dimensions of parenting attitude that did not have a significant impact on suicide risk.

**Table 2.** Differences in key variables by OSY status (*N* = 6,689)

| Variables | | Total | | SYs (*N* = 5,937) | | OSYs (*N* = 752) | | |
|---|---|---|---|---|---|---|---|---|
| | | M | SD | M | SD | M | SD | *t*(sig.) |
| Parenting attitude | Warmth | 3.40 | .68 | 3.44 | .66 | 3.15 | .80 | 9.237*** |
| | Autonomy support | 3.40 | .66 | 3.42 | .64 | 3.19 | .78 | 7.876*** |
| | Structure | 3.04 | .70 | 3.07 | .68 | 2.83 | .80 | 7.752*** |
| | Rejection | 1.56 | .64 | 1.54 | .62 | 1.71 | .73 | −5.895*** |
| | Coercion | 1.91 | .70 | 1.90 | .69 | 1.99 | .77 | −2.980** |
| | Chaos | 1.80 | .69 | 1.78 | .68 | 1.96 | .78 | −6.089*** |
| Suicide risk | | .16 | .50 | .13 | .42 | .46 | .87 | −10.266*** |

**p < .01,
***p < .001.

Furthermore, the results of the analysis showed that OSY status, a moderator variable, did not have a significant effect on the suicide risk. In the case of interaction terms, structure × OSY status (Coef. = −.108, $p < .01$), rejection × OSY status (Coef. = .152, $p < .001$) and chaos × OSY status (Coef. = .099, $p < .01$) had a significant effect on the suicide risk. That is, among the types of parenting attitudes, the moderating effect of OSY status was confirmed to be significant in the relationships among structure, rejection, chaos and suicide risk. On the other hand, for the other types of parenting attitude, the moderating effect of OSY status was confirmed to be non-significant, as can be seen in the relationships among warmth, autonomy support, coercion and suicide risk.

The specific trend in the moderating effect of OSY status in the relationship between structure among parenting attitude and suicide risk is presented in Figure 2. The graphical presentation confirmed that the suicide risk decreased more rapidly with increasing structure in the case of OSYs than in SYs.

The specific trend in the moderating effect of OSY status in the relationship between rejection among parenting attitudes and suicide risk is presented in Figure 3. The graphical presentation confirmed that the suicide risk increased more rapidly with increasing rejection in the case of OSYs than in SYs.

The specific trend in the moderating effect of OSY status in the relationship between chaos among parenting attitudes and suicide risk is presented in Figure 4. The graphical presentation confirmed that the suicide risk increased more rapidly with increasing rejection in the case of OSYs than in SYs.

## Conclusion and suggestions

The purpose of this study was to examine and verify the moderating effect of OSY status in the effect of parenting attitudes on the suicide risk of adolescents. To this end, a total of 5,937 SYs and 752 OSYs were analysed using the data from the Mental Health Survey of Korean Adolescents conducted by the National Youth Policy Institute.

The analysis results were as follows: First, in terms of the differences in key variables, among types of parenting attitudes,

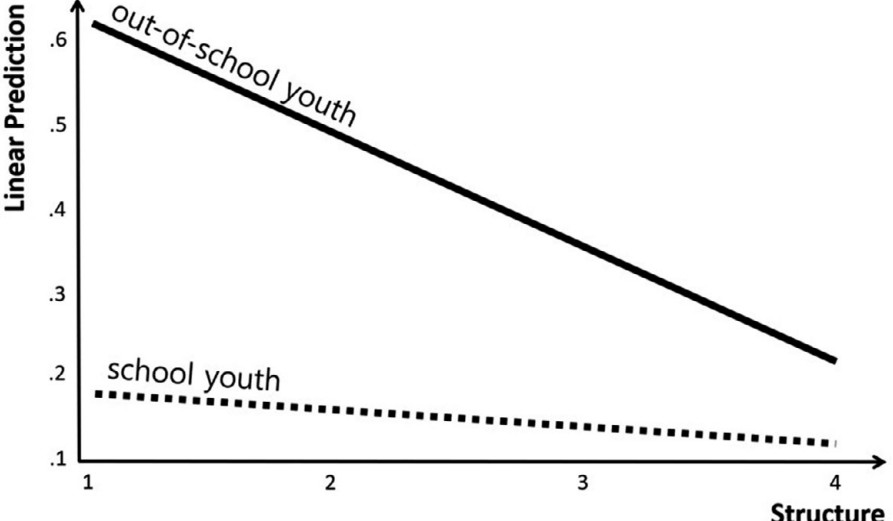

**Figure 2.** Analysis of interactions: structure × out-of-school youth status.

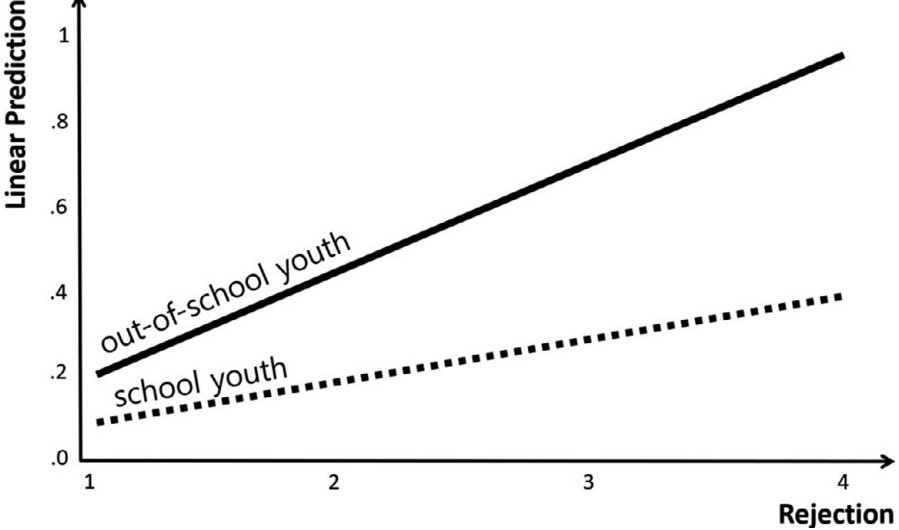

**Figure 3.** Analysis of interactions: rejection × out-of-school youth status.

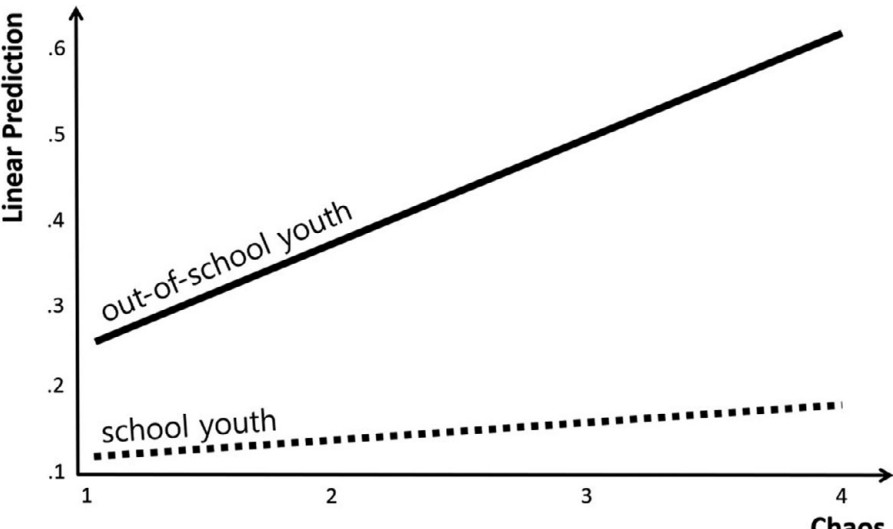

**Figure 4.** Analysis of interactions: chaos × out-of-school youth status.

warmth, autonomy support and structure were significantly higher in SYs, while rejection, coercion, chaos and suicide risk were significantly higher in OSYs. Previous studies using the same scale as this study also reported that negative parenting attitude has a direct effect on juvenile delinquency (Baek, 2022; Kim et al., 2023). Moreover, the results indicated that there is a necessity for differentiated measures of interventions for the OSY population, as they have a higher likelihood of exposure to suicide risk factors (Chung et al., 2010; Lee and Lee, 2023).

Second, in the relationship between key variables and suicide risk, female adolescents had a higher suicide risk than male adolescents. The results support the findings of previous studies showing that female adolescents are more vulnerable to the problem of suicide (Beautrais et al., 2006; Kim and Hong, 2012). In terms of the dimensions of parenting attitude, the lower the warmth, the lower the autonomy support, the higher the rejection and the higher the suicide risk of adolescents. This finding supports the reports from some of the previous studies that emphasized the importance of parenting attitude as a predictor of suicide risk (Wright, 1985; Hollis, 1996; Kim and Han, 2014; Choi and Cho, 2016; Choi and Kim, 2019), which indicates that measures with emphasis on primary environmental conditions/factors need to be established.

Third, the moderating effect of OSY status was confirmed in the relationship between parenting attitude and suicide risk. Looking into the individual types of parenting attitudes, the higher the structure among parenting attitudes, the more the suicide risk of OSYs decreased compared to that of SYs. In addition, with the parenting attitudes of increasing rejection and chaos, the suicide risk of OSYs increased more rapidly than that of Sys. These results were significant in that the findings supplemented the limitations of previous studies that mainly focused on SYs (Kim et al., 2023; Lee and Lee, 2023) or those studies that analysed SYs and OSYs separately (Chung et al., 2010); they revealed specific factors that showed differences per the characteristics of SY and OSY populations.

The suggestions based on the above results of this study are as follows: First, programs for parenting attitude education need to be developed to reduce the suicide risk of adolescents. Considering that one of the reasons for the increase in the Korean adolescent suicide rate to 2.7 per 100,000 population involves factors of the home/family environment, such as parental attitudes, behaviour and thinking (Kim and Han, 2014; Korea Disease Control and Prevention Agency, 2022), development of education programs for parents will serve as an effective measure for primary prevention of adolescent suicide. From the results of this study, among the dimensions of parenting attitude, warmth, autonomy support and rejection were shown to be important predictors of suicide risk in adolescents. Thus, the significant contribution of this study lies in that analysis was made for sub-types of parenting attitudes, unlike the approaches in existing studies. That is, the findings indicate that it is necessary to implement a program that takes into account warmth, autonomy support and rejection in parenting attitude education for parents of adolescents at risk of suicide and to emphasize the direction of the appropriate parenting attitude. Recently, in Korea, policy support for crisis intervention programs for adolescents at high risk of suicide/self-harm has been expanded, such as an 'Intensive Psychological Clinic for High-risk Adolescents'. (Lee and Lee, 2023). Through a comprehensive psychological assessment by experts, measures for a rapid initial response and intervention for adolescents at high risk of suicide have been prepared, but counseling programs for parents and follow-up management services are still in the early stages (Kim and Han, 2014). Furthermore, not establishing systems for examining underlying factors to reduce the risk of adolescents' suicide can be viewed as a limitation, since previous education programs on the issue of adolescent suicide have mainly covered parenting attitudes such as abuse and neglect (Park, 2014). Therefore, the development of parent education programs enabling the enhancement of warmth and autonomy support and reduction of rejection among parenting attitude types is expected to have a positive effect on reducing suicide risk among adolescents.

Second, strategies and measures for suicide prevention intervention for OSYs need to be established at the governmental level. The results of this study showed that there was a moderating effect of OSY status in the relationship among structure, rejection, chaos and suicide risk, among parenting attitudes. OSYs, who spend most of their time at home, are not only relatively more influenced by their parents (Lee and Lee, 2023) but also are left with an environment with more difficulties in access to institutional education compared to SYs. Thus, different environmental characteristics

between different adolescent populations may show different patterns in terms of the expression of suicidal thoughts/behaviour; moreover, considering the suicide rate of OSYs not counted in official statistics, the actual rate of adolescent suicide may be even higher.

Recently, organizations and institutions related to adolescents are making attempts to run outreach programs to provide early interventions for OSYs are at risk of suicide, but guidelines for crisis responses have not been prepared. To this end, direct support and services for OSY suicide prevention should be linked at the governmental level, and guidelines for early intervention should be established. As shown in the results of this study, among the types of parenting attitudes structure, rejection and chaos were associated with the suicide risk of OSYs, and there can be a difference in the influence of parenting attitude factors on the suicide risk between SYs and OSYs. Therefore, the dissemination of education programs for parenting attitudes focusing on structure, rejection and chaos will provide direct and practical help in providing differentiated interventions for OSYs.

## Limitations of the study

Due to the nature of secondary data used in this study, there were limitations in using a variety of control variables, and age groups could not be categorized further. That is, in the data of the Mental Health Survey of Korean Adolescents used in this study, there were no variables other than gender, age and household income level that could be used as control variables. In addition, age was categorized only into middle school age or under and high school age. Accordingly, it is expected that in future follow-up studies, if more diverse control variables can be used and age data becomes available as a continuous variable, the results will provide clearer insights into a variety of factors or variables influencing adolescent suicide risk. In addition, since the exact size of out-of-school youth, one of the subjects of this study was unable to be estimated, it was difficult to identify the population, and thus, random sampling was unable to be conducted as with school youth. This inevitably led to limitations in the representativeness, bias and comparability of the sample. It was thus expected that follow-up studies would clearly identify the population of out-of-school youth so that there are no problems with the representativeness and comparison of the sample through random sampling.

**Open peer review.** To view the open peer review materials for this article, please visit http://doi.org/10.1017/gmh.2025.35.

**Data availability statement.** The data that support the findings of this study are openly available at https://www.nypi.re.kr/archive/mps

**Author contribution.** Kyu-Hyoung Jeong constructed the research model and wrote the research methods and results Sung-Hee Lee was responsible for the translation and final review of the paper. A-Ran Park wrote the conclusions based on the findings Do-Hun Song wrote the introduction and edited the manuscript. All authors reviewed and approved the final version of the manuscript.

**Financial support.** This paper was carried out with support from the Ministry of Education, Republic of Korea, and the National Research Foundation of Korea (NRF), BK21 Future Welfare Developing Human Resources for the Community Innovation (4199990314436).

**Competing interests.** The authors declare none.

**Ethics statement.** All methods were performed in accordance with the Declaration of Helsinki. This report was exempted from approval by the institutional review boards (IRB) of the Clinical Research Ethics Committee of Jeonbuck national University (IRB number: JBNU 2024–2103-021). Every participant gave a written consent prior to their participation in the study.

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
