## [Reviewer Report]

1.I am not sure if there are any regulations regarding the journal, but would a structured abstract, including sections such as background, methodology, results, and conclusions, be clearer and more concise?

2.Does the evidence support the author’s statement in the introduction that “South Korea (henceforth ‘Korea’), the country with the highest rate of suicide among the Organization for Economic Cooperation and Development (OECD) member nations, shows a worrisome development in the problem of adolescent suicide?”

Additionally, the author mentions in the introduction that, “Furthermore, for school-age adolescents, school is generally the place where they spend most of their time during the day, and along with academic knowledge, they acquire skills of socialization through interpersonal relationships with different people, which cannot be learned at home.” Would it be possible to include 1 to 2 references to substantiate this point?

3.Could the author provide additional theoretical support and a conceptual framework for this study?

4.The author mentions in the data section that “The SYs were extracted by stratified cluster sampling, and OSYs by convenience sampling” . How do these two different sampling methods ensure the representativeness and comparability of the samples when targeting two different subject populations?

5.The author categorizes the household income level into three classes: very low, average, and very high, with the values assigned as very low = 1, average = 4, and very high = 7. What are the criteria and basis for this classification? What is the significance of this numerical assignment?

6.Where is the reference to Table 1 in the manuscript?

7.The description of the results in Table 2 is insufficiently clear, specific, and relatively difficult to comprehend.

8.The author took Out-of-School Youth (OSY) as a moderator variable, and further clarification is needed on whether it moderates parenting attitudes or suicide risk.

Table 3 indicates that there are significant statistical differences in six types of parenting attitudes and suicide risk between SY and OSY. Is it appropriate to use OSY as a moderator variable? Are there any biases in the results when conducting research on the moderating effects?

In addition, in the results, the author only examined the interaction between parenting attitudes and OSY. How are the moderating effects reflected?

---

## [Reviewer Report]

Appreciate the authors for working on a clinically relevant area. Overall, the work and the manuscript look impressive.

Title looks fine. Abstract conveys the summary of the study.

Introduction: Background and rationale has been stated well. Objective may be specifically stated at the end of the introduction.

Methods: Kindly mention the study design. Participants, variables and statistical methods have been described. How was adolescents defined for the study. Please include.

Please mention whether informed consent was taken.

Results: has been elaborated well. Participants, Descriptive data, Outcome data, and moderating effects have been included.

Discussion and interpretation of results have been provided.

Development of education programs for patients will serve as the measure for primary prevention of adolescent suicide (Line 266).Please clarify whether “parents” instead of patients.

Limitations of the study written in detail.

Funding: has been mentioned.

Kindly mention conflict of interest, if any.

Corresponding author has not been mentioned.

The manuscript may benefit from a statistical review.

---

## [Reviewer Report]

I find the paper well written. Objectives are clear, and methods are in congruency. The statistics are simple; yet, adequate.

The use of tables and figures are helpful to follow results and discussion.

---

## [Editor Report]

The work is interesting and the problem is also relevant. The authors must answer some questions from the reviewers, before possible publication in this journal: 

1. The author mentions in the introduction that, “Furthermore, for school-age adolescents, school is generally the place where they spend most of their time during the day, and along with academic knowledge, they acquire skills of socialization through interpersonal relationships with different people, which cannot be learned at home.” Would it be possible to include 1 to 2 references to substantiate this point?

2. Could the author provide additional theoretical support and a conceptual framework for this study?

3. The author mentions in the data section that “The SYs were extracted by stratified cluster sampling, and OSYs by convenience sampling” . How do these two different sampling methods ensure the representativeness and comparability of the samples when targeting two different subject populations?

4. The author categorizes the household income level into three classes: very low, average, and very high, with the values assigned as very low = 1, average = 4, and very high = 7. What are the criteria and basis for this classification? What is the significance of this numerical assignment?

5. Add the reference to Table 1 within the manuscript.

6. The description of the results in Table 2 needs to be made more detailed and clear. Please review the way they are described.

7. The author took Out-of-School Youth (OSY) as a moderator variable, and further clarification is needed on whether it moderates parenting attitudes or suicide risk.

8. Table 3 indicates that there are significant statistical differences in six types of parenting attitudes and suicide risk between SY and OSY. Is it appropriate to use OSY as a moderator variable? Are there any biases in the results when conducting research on the moderating effects?

9. In addition, in the results, the author only examined the interaction between parenting attitudes and OSY. How are the moderating effects reflected? 

10. The objective could be described at the end of the Introduction.